# Central Carbon Metabolism in *Candida albicans* Biofilms Is Altered by Dimethyl Sulfoxide

**DOI:** 10.3390/jof10050337

**Published:** 2024-05-08

**Authors:** Maria Fernanda Cordeiro Arruda, Romeu Cassiano Pucci da Silva Ramos, Nicoly Subtil de Oliveira, Rosimeire Takaki Rosa, Patrícia Maria Stuelp-Campelo, Luiz Fernando Bianchini, Silas Granato Villas-Bôas, Edvaldo Antonio Ribeiro Rosa

**Affiliations:** 1Graduate Program on Dentistry, School of Medicine and Life Sciences, Pontifical Catholic University of Paraná, Curitiba 80215-901, Brazil; mafercordeiro@hotmail.com (M.F.C.A.); romeucs@gmail.com (R.C.P.d.S.R.); 2Graduate Program on Animal Sciences, School of Medicine and Life Sciences, Pontifical Catholic University of Paraná, Curitiba 80215-901, Brazil; nicolysubtil@gmail.com; 3Xenobiotics Research Unit, School of Medicine and Life Sciences, Pontifical Catholic University of Paraná, Curitiba 80215-901, Brazil; rosimeire.rosa@pucpr.br (R.T.R.); p.campelo@pucpr.br (P.M.S.-C.); fernando.bianchini@pucpr.br (L.F.B.); 4Luxembourg Institute of Science and Technology (LIST), 4940 Hautcharage, Luxembourg; silasgvboas@gmail.com

**Keywords:** *Candida albicans*, biofilm, dimethyl sulfoxide, metabolic pathways, metabolomics

## Abstract

The effect of dimethyl sulfoxide (DMSO) on fungal metabolism has not been well studied. This study aimed to evaluate, by metabolomics, the impact of DMSO on the central carbon metabolism of *Candida albicans*. Biofilms of *C. albicans* SC5314 were grown on paper discs, using minimum mineral (MM) medium, in a dynamic continuous flow system. The two experimental conditions were control and 0.03% DMSO (*v*/*v*). After 72 h of incubation (37 °C), the biofilms were collected and the metabolites were extracted. The extracted metabolites were subjected to gas chromatography–mass spectrometry (GC/MS). The experiment was conducted using five replicates on three independent occasions. The GC/MS analysis identified 88 compounds. Among the 88 compounds, the levels of 27 compounds were markedly different between the two groups. The DMSO group exhibited enhanced levels of putrescine and glutathione and decreased levels of methionine and lysine. Additionally, the DMSO group exhibited alterations in 13 metabolic pathways involved in primary and secondary cellular metabolism. Among the 13 altered pathways, seven were downregulated and six were upregulated in the DMSO group. These results indicated a differential intracellular metabolic profile between the untreated and DMSO-treated biofilms. Hence, DMSO was demonstrated to affect the metabolic pathways of *C. albicans*. These results suggest that DMSO may influence the results of laboratory tests when it is used as a solvent. Hence, the use of DMSO as a solvent must be carefully considered in drug research, as the effect of the researched drugs may not be reliably translated into clinical practice.

## 1. Introduction

There are about 46,000 cases of hospital-acquired *Candida* spp. infection and 220 related deaths reported each year in the United States of America. These hospital-acquired infections cause a severe economic loss annually [1]. Among the members of the *Candida* genus, *Candida albicans* is the most important species that causes severe systemic infections. Infections of *C. albicans* are challenging to treat as they form biofilms and develop resistance against commonly used antifungals [1,2,3]. *Candida albicans* is used as a model to investigate and validate the effect of novel antifungal drugs. Additionally, *C. albicans* is a valuable resource to understand the cellular mechanisms underlying antifungal resistance, fungal virulence, and fungi’s susceptibility to antifungal agents [4].

To ensure experimental reproducibility and pharmacological safety, conventional antifungal agents, predominantly hydrophobic, are dissolved in aprotic solvents exhibiting low toxicity and high polarity to validate their antifungal activity [5]. Dimethyl sulfoxide (DMSO) is widely used as a solvent for antifungal agents in drug susceptibility assays. The final concentration of DMSO used in the test medium is about 0.03% (*v*/*v*) [5,6]. DMSO is also used in the cryopreservation of cells at a concentration of about 10% (*v*/*v*) [7].

DMSO has been used in the laboratory since the 1950s [8]. Although DMSO has several laboratory applications, its effect on the cellular metabolism of yeast cells is not entirely understood.

It is known that DMSO can inhibit some *Candida* species’ growth at concentrations lower than 1% (*v*/*v*) [9]. At concentrations ranging from 2.5–7.5% (*v*/*v*), it dose-dependently inhibits the formation of *C. albicans* germ tubes [10]. Therefore, higher concentrations of DMSO may compromise cell viability and result in cell death, as observed in *Saccharomyces cerevisiae*, which also belongs to the Saccharomycetales order [11]. Several studies have demonstrated that a treatment with DMSO results in cellular morphological changes, which affect the validity of the results in antifungal susceptibility tests [12,13,14,15].

The toxicity is mainly due to its affinity with the polar region of the lipid bilayer. The interaction between DMSO and the polar region results in the disorganisation of lipids and enhanced membrane permeability [16,17,18,19,20]. This characteristic is helpful in pharmacological and genetic studies, as enhanced membrane permeability facilitates the absorption of antifungal agents and the incorporation of DNA [21,22]. Additionally, DMSO is used for the cryopreservation of cells as it interrupts the network of water molecules near the surface of the lipid membrane, which reduces the stress caused by changes in the volume of water during the freezing and thawing process [23]. At concentrations higher than 10%, DMSO is reported to damage the integrity of the yeast cell membrane [24]. However, DMSO’s effect on the cells’ metabolic changes has not been well studied.

It was shown that three cellular processes are required for solvent resistance to DMSO in *S. cerevisiae*: (1) transport between the Golgi complex and endoplasmic reticulum; (2) the action of the SWR1 complex, a protein complex involved in chromatin remodelling and stress responses, including osmotic and cell wall stress; and (3) DNA repair, which differs from the toxicity mechanisms observed in nematode and human cells [25].

Studies on *S. cerevisiae* have demonstrated that 1% DMSO affects the expression of 1338 of its genes, downregulates the lipid biosynthetic pathways, and upregulates the amino acid biosynthetic pathway [26]. DMSO affects the expression of genes involved in environmental stress responses [27]. Zhang et al. [28] demonstrated that DMSO affects the expression of 73 *S. cerevisiae* genes, including genes involved in metabolising carbohydrates, amino acids, nucleotides, lipids, and phosphate. These studies, however, only evaluated the impacts of DMSO on the expression of some genes associated with metabolic pathways. To our knowledge, no metabolic-wide landscape evaluations have been carried out.

Regarding functional genomics, metabolomics is a post-genomic tool for understanding intracellular and extracellular metabolic processes. Metabolomic studies facilitate predicting a metabolic response to a test substance [29]. There is a growing interest in evaluating the effect of DMSO on cellular metabolism as DMSO is used as a solvent for new drugs. This study aimed to assess the impact of DMSO on the metabolic profile of *C. albicans* biofilms using metabolomics.

In this study, we evaluate the qualitative and quantitative differences among the intracellular metabolites of *C. albicans* biofilms grown only with culture medium and in the presence of a low DMSO concentration. We found that the DMSO altered 13 metabolic pathways involved in primary and secondary cellular metabolism, using metabolomics-based approaches. Seven pathways were downregulated and six were upregulated in the DMSO group.

## 2. Materials and Methods

### 2.1. Candida albicans Strain

The *Candida albicans* SC5314 strain was cultured in a medium containing 1% yeast extract, 1% peptone, 2% dextrose, and 2% agar (YPDA, pH 7) at 32 °C.

### 2.2. Preparation of Inoculum

The inoculum was prepared by incubating a colony of *C. albicans* in minimum mineral [MM; per litre: glucose 10 g; (NH_4_)_2_SO_4_ 2 g; KH_2_PO_4_ 2 g; CaCl_2_.2H_2_O 0.05 g; MgSO_4_.7H_2_O 0.05 g; ZnSO_4_.7H_2_O 0.05 g; CuSO_4_.5H_2_O 0.01 g; FeSO_4_.7H_2_O 0.1 g; final pH 6.5] medium at 125 rpm and 30 °C for 18 h (exponential phase) in the absence or presence of 0.03% DMSO (*v*/*v*). The absorbance of the cell suspension at 600 nm (OD_600 nm_) was adjusted to 1.0 (3 × 10^7^ cells mL^−1^). The cells were washed and resuspended in sterile 145 mM NaCl.

### 2.3. Experimental Groups

The experiment was divided into two groups consisting of five replicates, carried out on three independent occasions.

(a)Control group: cells were cultured in standard MM medium (replicates: C1a, C2a, C3a, C4a, C5a, C1b, C2b, C3b, C4b, C5b, C1c, C2c, C3c, C4c, and C5c).(b)DMSO group: cells were cultured in MM medium supplemented with 0.03% (*v*/*v*) DMSO (replicates: D1a, D2a, D3a, D4a, D5a, D1b, D2b, D3b, D4b, D5b, D1c, D2c, D3c, D4c, and D5c).

### 2.4. Formation of Aerobic Biofilms under Continuous Flow

The antibiogram paper discs (five discs/biological replicate) were dried at 60 °C for 24 h and incubated overnight in a desiccator with P_4_O_10_, under vacuum conditions. The mass of the discs was determined before the discs were autoclaved. A 10 µL aliquot of the *C. albicans* SC5314 suspension was inoculated onto the discs. The inoculated discs were placed into a paper biofilm growth reactor (PEBR), as described by Selow et al. [30].

The discs were drip irrigated with MM medium in the presence or absence of 0.03% DMSO (*v*/*v*). A continuous flow rate of 30 mL h^−1^ was established for 72 h at 37 °C. Each experiment had a “white” sample, in which the discs were not inoculated with the yeast cells.

### 2.5. Extraction of Intracellular Metabolites

Metabolomic analysis was performed according to the protocol described by Smart et al. [31], with minor modifications. The intracellular metabolites (fingerprint) were carefully extracted under low temperatures to minimise the loss or degradation of these substances by heat.

### 2.6. Sampling and Quenching (Fixation of Cellular Metabolism)

The discs containing the mature *C. albicans* biofilms were collected by sonication for 3 min in an ultrasonic bath sonicator (USR 30H; Merck Eurolab, Darmstadt, Germany), at a frequency of 35 kHz to disrupt the biofilm, and quickly centrifuged (15,000× *g*; 4 °C; 10 min) and inserted into Nalgene^®^ (Rochester, NY, USA) tubes containing 50% (*v*/*v*) frozen methanol. A 20 µL aliquot of the internal standard d4-alanine (2,3,3,3-d4-alanine) was added to each sample. The tubes were capped and incubated in an ice-cold ethanol bath (−23 °C) for 5 min. The tubes were vortexed for 1 min and conditioned at −80 °C for 30 min.

### 2.7. Extraction and Quantification of Intracellular Metabolites

The cell membrane was ruptured by three freeze–thaw cycles to extract the metabolites. The samples were thawed in an ice bath for 5 min, vortexed for 1 min, and frozen at −80 °C for 40 min. The lysates were centrifuged at −20 °C and 30,600× *g* for 20 min. The supernatant was collected and stored at −80 °C. The precipitate containing the cellular contents and discs was resuspended in 70% (*v*/*v*) cold methanol by vortexing for 1 min. Next, the samples were centrifuged at −20 °C and 30,600× *g* for 20 min. The supernatant was collected and added to 20 mL of cold Milli-Q^®^ (Paraná, Brazil) water (4 °C). The samples were frozen at −80 °C. The frozen sample was subjected to freeze drying (−83 °C). The freeze-dried samples were stored at −80 °C.

### 2.8. Quantification of Cell Biomass

The cell biomass was quantified by resuspending the sample containing the precipitate of the cellular contents and discs in 10 mL of distilled water. The sample was filtered and washed using a Millipore^®^ filtration system (Merk KGaA, Darmstadt, Germany). The samples were filtered through a membrane (0.22 µm) previously dried at 60 °C for 24 h and stored overnight in a desiccator containing P_4_O_10_, under vacuum conditions. The mass of the membrane was determined. The filtering membranes containing the residues were dried as described previously. The biomass values were determined by subtracting the value of the membrane and disc masses before filtration from the value of the mass of the membrane and discs containing cell debris.

### 2.9. Derivatization with Methyl Chloroformate (MCF)

MCF derivatisation involves alkylation, which converts amino acids and carboxylic acids into carbamates and volatile esters. This enables the identification of primary cellular metabolites, including most central carbon metabolism intermediates.

Vortexing resuspended the lyophilised samples in 200 µL of 1 M NaOH. The samples were transferred to silanised tubes containing 167 µL of methanol and 34 µL of pyridine. The samples were positioned on the vortex, and 20 µL of MCF was added and incubated for 30 s. Next, 20 µL of MCF was added and incubated for 30 s, followed by 400 µL of chloroform. The samples were incubated on a shaker for more than 10 s. Further, 400 µL of 50 mM NaHCO_3_ was added to the samples and vortexed for 10 s. The samples were centrifuged for 3 min at 1500× *g*. The aqueous phase was removed, and 100 mg of anhydrous Na_2_SO_4_ was added to the chloroform phase to remove any remaining water. The samples were transferred to vials for gas chromatography.

### 2.10. Gas Chromatography and Mass Spectrometry

Gas chromatography–mass spectrometry (GC/MS) analysis was performed using a Shimadzu^®^ GCMS-QP2010 system equipped with a selective quadrupole mass detector in its electronic impact operation mode (70 eV) and a ZB1701 column (Zebron, Phenomenex^®^, Torrance, CA, USA) of 30 m × 250 µm × 0.15 µm. The mass spectrometer was operated in the scan mode (beginning after 4.5 min, mass range 40–650 amu at 0.15 s/scan). The parameters for the separation and analysis of the products derived using MCF are described elsewhere [31].

### 2.11. Statistical Analysis

The metabolites in the samples were identified based on the mass spectra and retention times generated by GC/MS using the AMDIS (Automated Mass Spectral Deconvolution and Identification System; http://chemdata.nist.gov/mass-spc/amdis (aceessed on 31 January 2024)). The data were compared with the library of mass spectra provided by the University of Auckland (New Zealand). The relative abundance of the identified metabolites was determined using the ChemStation A.10.02 software (Agilent^®^, Santa Clara, CA, USA).

The raw data were normalised to the internal standard d4-alanine, “white”, and biomass. Low-abundance compounds were excluded. Student’s *t*-test (*p* < 0.05) and principal component analysis (PCA) were performed using the statistical software R 4.2.3 (R Foundation, London, UK) to remove the outliers. PAPi (Pathway Activity Profiling) [32] followed by Students’ *t*-test (*p* < 0.05) and a PCA were used to predict the metabolic pathways. The metabolic pathways were analysed using the KEGG (Kyoto Encyclopedia of Genes and Genomes; https://www.genome.jp/kegg/), BioCyc (http://biocyc.org/), and CGD (Candida Genome Database; http://www.candidagenome.org/) online databases.

## 3. Results

*Candida albicans* biofilms were cultivated in a paper biofilm growth reactor (PEBR) [30], with the absence or presence of DMSO [0.03%]. After 72 h of growth, the biofilms were collected and processed using the method described by Smart et al. [31] to obtain their intracellular metabolites, which were analysed by GS-MS and statistically compared.

### 3.1. Principal Component Analysis (PCA)

A PCA was performed using the mass spectra data and relative abundance of the substances present. The result showed that the samples were arranged in the control and DMSO groups distinctively based on the abundance of the present substances. 

### 3.2. Metabolite Analysis

The GC-MS analysis revealed 88 intracellular metabolites in the *C. albicans* SC5314 biofilms (Table 1). Among the 88 intracellular metabolites, 27 metabolites exhibited differential abundance (*p* < 0.05) between the control and DMSO groups (Table 2). Among these 27 metabolites, the DMSO group exhibited decreased levels of eight metabolites and increased levels of 19 (Figure 1, Figure 2 and Figure 3). There were no quantitative differences in the biomasses of the biofilms from the two groups.

The DMSO group exhibited lower levels of the tyrosine, threonine, ornithine, lysine, methionine, and histidine amino acids than the control group. Additionally, the DMSO group showed decreased levels of gamma-aminobutyric acid (GABA) and carbamic acid, an unstable amine.

DMSO enhanced the levels of amino acid derivatives, such as pyroglutamic acid, N-acetylglutamic acid, putrescine, and S-adenosylmethionine. DMSO enhanced tricarboxylic acid cycle (TCA) intermediates such as fumaric acid, malic acid, alpha-ketobutyric acid, and alpha-ketoglutaric acid. The DMSO group exhibited enhanced levels of dehydroascorbic acid (the oxidised form of ascorbic acid), glutathione cofactor, and fatty acids, such as 10,13-dimethyltetradecanoic acid (10,13-DMTDA), dodecanoic acid, myristic acid, and stearic acid. Furthermore, the DMSO group exhibited enhanced levels of itaconic acid (an intermediate in the metabolism of C5 branched chain dibasic acids), 1-aminocyclopropane-1-carboxylic acid (ACC) (an intermediate in the metabolism of methionine), dehydroabietic acid (an isoprenoid), benzoic acid (an aromatic carboxylic acid), and malonic acid (an inhibitor of cellular respiration).

### 3.3. Pathway Activity Profiling (PAPi) Analysis

PAPi analysis was used to visualise the effect of DMSO on metabolic pathways. The metabolic pathway analysis revealed that DMSO altered 13 primary and secondary cellular metabolism pathways. Seven pathways were downregulated in the DMSO group and six were upregulated (Figure 4).

The downregulated metabolic pathways in the DMSO group were associated with fatty acid biosynthesis and carbohydrate metabolism. The upregulated pathways were mainly involved in the metabolism of amino acids and cofactors/vitamins (Table 3).

Figure 1 shows the metabolic network of the DMSO group, illustrating the major altered metabolic pathways and the concentrations of their respective metabolites in the *C. albicans* SC5314 biofilms.

## 4. Discussion

This study evaluated the effect of DMSO on cellular metabolism at a low concentration (0.03%), as DMSO is diluted to this concentration in antimicrobial susceptibility assays [5].

In an attempt to detoxify DMSO to dimethyl sulfide (DMS), *S. cerevisiae* catalyses this reduction using methionine sulfoxide reductase (Msr), which is encoded by the *MXR1* gene [33,34]. However, DMSO was reported to selectively inhibit the antioxidant function of one of the three Msr allotypes, resulting in enhanced oxidative stress sensitivity [34]. In some yeasts, DMSO can be metabolised to DMS by aryl-alcohol dehydrogenase and subsequently to S-adenosylmethionine in the methionine pathway. S-adenosylmethionine is used in methylation reactions during phospholipid synthesis [35]. The *MXR1*, *IFD6,* and *LPG20* genes in *Candida albicans* SC5314 encode enzymes like aryl-alcohol dehydrogenase [36]. In this study, the DMSO group exhibited enhanced levels of intracellular S-adenosylmethionine. However, the biosynthesis of phospholipids was unaffected in the DMSO group. The enhanced levels of S-adenosylmethionine are compatible with DMSO metabolism, which was not detected in our samples.

Generally, higher concentrations of intracellular metabolites indicate a decreased metabolic flux toward their respective pathways. Conversely, reduced concentrations of metabolites indicate the upregulation of their respective pathways [36].

### 4.1. Lipid Metabolism

Of the downregulated pathways in the DMSO group, the fatty acid biosynthetic pathway was markedly changed, followed by the carbohydrate metabolic pathway.

In yeasts, fatty acids can be incorporated into the phospholipids and sphingolipids or serve as an energy reserve in the form of triacylglycerols and sterol esters. Additionally, fatty acids function as transcriptional regulators, signalling molecules, and protein modifiers during the post-translational phase [37]. Lipids also play an important role in the formation of biofilms and antifungal resistance in *C. albicans* [38,39].

Therefore, we postulated that the decreased biosynthesis of fatty acids observed in the DMSO group might be due to the intracellular accumulation of 10,13-dimethyltetradecanoic acid (10,13-DMTDA), dodecanoic acid, myristic acid, and stearic acid, which were not utilised by the cells to prevent lipotoxicity [40,41]. Furthermore, the downregulation of the fatty acid biosynthetic pathway by DMSO may also be due to the downregulation of the pentose phosphate pathway and the decreased metabolism of pyruvate, which are essential for the biosynthesis of fatty acids.

DMSO enhances phospholipid biosynthesis by upregulating the expression of its associated genes in *S. cerevisiae*. These genes are involved in the repair of DMSO-mediated cell membrane damage [24]. Additionally, DMSO enhances the expression of the genes involved in methionine synthesis (not observed in this study) and the synthesis of cell wall components. León-García et al. [42] reported that DMSO dose-dependently enhanced the expression of cell wall proteins in *C. albicans*. These proteins are also involved in stress responses. However, we used a concentration of DMSO that was about 45 times lower than that used in other studies. Hence, we did not observe the repair of DMSO-mediated membrane damage.

The results of this study concur with those of studies that reported DMSO-mediated changes in the lipid composition of yeast cell membranes [26,43,44]. We believe that DMSO, at concentrations above those tested in this study, may lead to the sensitisation and reorganisation of membranes.

### 4.2. Carbon and Carbohydrate Metabolism

Unlike *S. cerevisiae*, *C. albicans* initiates fermentation only under anaerobic conditions (Crabtree-negative) and metabolises carbohydrates by oxidative phosphorylation [45]. Although the regulation of carbon assimilation differs between these species, the pathways of central carbon metabolism are identical among different yeast species [46,47]. Additionally, the metabolic responses to DMSO are similar between *S. cerevisiae* and *C. albicans*.

Zhang et al. [26] demonstrated that DMSO alters the expression of several genes involved in the central carbohydrate metabolism pathways (such as glycolysis and gluconeogenesis), TCA cycle, sugar metabolism pathway, and pentose phosphate pathway in *S. cerevisiae*. Also, the authors showed that DMSO decreased the levels of isoenzymes involved in glycolysis and enhanced the levels of the isoforms of pyruvate carboxylase, citrate synthetase, and isocitrate dehydrogenase, which are involved in the TCA cycle. This was similar to the observations of this study, where the DMSO treatment downregulated the pathways of pyruvate metabolism, pentose phosphate, interconversion between pentose and glucuronate, and ascorbate/aldarate metabolism. Additionally, DMSO enhanced the concentrations of the TCA cycle intermediates (α-ketoglutaric acid, fumaric acid, and malic acid).

In yeasts, the pyruvate formed during glycolysis can be processed in two different ways: (1) it can be directly converted to acetyl-coenzyme A by the pyruvate dehydrogenase (PDH) enzymatic complex after its transport into the mitochondria, or (2) it can be converted to acetyl-CoA in the cytosol by the PDH-bypass complex, usually under hypoxic conditions. In *C. albicans*, the expression of the genes involved in glycolysis and fermentation is upregulated under hypoxic conditions [47,48,49]. It has been demonstrated that pyruvate fermentation, via a PDH bypass, enhances acetaldehyde production in *C. albicans* under hypoxic and anoxic conditions and in the presence of glucose [50]. These results concurred with those that demonstrated that *C. albicans* is a weak ethanol producer [51].

In this study, the formation of mature membrane-based biofilms may have resulted in decreased oxygen flow to the inner layers of the biofilm. The oxygen supply can also be reduced due to the system used to grow biofilms (gas exchanges were not measured). Under these hypoxic conditions, the PDH bypass complex may be activated, which promotes the synthesis of acetyl-CoA and, subsequently, the synthesis of fatty acids, as seen in both experimental groups. However, several studies have reported that the mitochondrial PDH complex has a higher affinity for pyruvate than the PDH bypass complex at low glucose levels [52]. Thus, the DMSO group exhibited downregulated carbohydrate metabolism pathways (the pentose phosphate pathway and interconversion between pentose and glucose), consequently reducing pyruvate metabolism. We believe that the reduction in fatty acid biosynthesis in the DMSO group may be because the pyruvate metabolism has been directed primarily to the mitochondrial PDH complex. We postulated that enhanced levels of intracellular fermentation products (acetate and ethanol) may activate the glyoxylate cycle, which allows the cells to utilise these compounds as a carbon source [53]. This also explains the enhanced malic acid levels observed in the DMSO group.

### 4.3. Amino Acid Metabolism

DMSO upregulates the expression of most amino acid biosynthetic isoenzymes, such as aspartate-semialdehyde dehydrogenase (hom2), which are involved in the biosynthesis of lysine, methionine, and threonine in *S. cerevisiae* [26]. This was not consistent with the phenotype observed in this study. Zhang et al. [26] also reported that DMSO enhanced the expression of five enzymes involved in histidine biosynthesis. However, the DMSO group exhibited a decreased expression of such enzymes, possibly due to the enhanced metabolism of *C. albicans*. Furthermore, DMSO was reported to enhance the expression of enzymes involved in the urea cycle and, consequently, the arginine biosynthesis pathway, which concurred with the observations of this study.

The upregulation of arginine metabolism was reported during the morphogenesis of yeast to a filamentous form in *C. albicans*. The upregulation of arginine metabolism observed in this study may be due to filamentation, which usually occurs in response to environmental changes as an adaptation and survival strategy against xenobiotics [54,55,56,57]. Jiménez-López et al. [58] reported that the genes involved in arginine biosynthesis are induced in *C. albicans* even under the oxidative stress conditions prevalent during phagocytosis.

Polyamines are reported to regulate hyphae formation in *C. albicans* [59,60] and are required for meiosis and sporulation [61]. Therefore, enhanced levels of ornithine and polyamine indicate a yeast-to-hyphae transition, which protects the fungus. Cao et al. [62] demonstrated that an exposure to amphotericin B enhances the levels of polyamines in microorganisms. In this study, enhanced levels of putrescine (a precursor of polyamines) and decreased levels of ornithine were observed after an exposure to DMSO.

Kamthan et al. [63] observed a marked increase in the levels of lysine and tyrosine during the morphogenesis of yeast to its filament form, induced by n-acetylglucosamine. However, we demonstrated that DMSO decreases the levels of these amino acids. Tyrosine is a precursor of tyrosol, a quorum-sensing secretory molecule that stimulates filamentation and biofilm formation in *C. albicans*. A DMSO treatment decreased the levels of tyrosine, which may be due to the production and excretion of tyrosol (unaltered intracellular levels). Additionally, DMSO may upregulate filamentation signalling [64,65].

Some findings in this study did not concur with those observed in other studies. DMSO was reported to enhance tyrosine levels and upregulate fatty acid metabolism during morphological transitions [57]. However, it is essential to note that most studies evaluated the yeast-to-hypha transition and not the filamentation in mature biofilms assessed in this study. Several studies have demonstrated differential cytoplasmic protein profiles between yeast cells, hyphae, and biofilms [66].

GABA is a non-proteinogenic amino acid synthesised by the decarboxylation of glutamic acid or by the degradation of polyamines [67]. DMSO reduces the levels of GABA, which enhances putrescine levels. Although GABA is reported to be a nitrogen source for *S. cerevisiae* [68], Coleman et al. [69] proposed that intracellular GABA protects *S. cerevisiae* against oxidative stress. The GABA degradation pathway converts GABA into succinic acid in *S. cerevisiae*. This pathway also appears to be present in *C. albicans,* as related orthologous genes are observed in both species [70].

The upregulation of beta-alanine metabolism, which is essential for the formation of coenzyme A, may be due to the enhanced levels of S-adenosyl methionine and putrescine (and consequently decreased levels of methionine), which combine to form spermidine, a precursor of beta-alanine. Additionally, the enhanced levels of S-adenosyl methionine and putrescine may have resulted from enhanced malonic acid levels. Malonic acid can form malonate semialdehyde, which is involved in beta-alanine biosynthesis. However, malonic acid is known to be a competitive inhibitor of succinate dehydrogenase, an enzyme involved in the TCA cycle and electron transport chain that reduces cellular respiration [71]. The enhanced levels of malic acid observed after the treatment with DMSO at a concentration of 0.03% (*v*/*v*) may be due to the activation of the glyoxylate cycle. The glyoxylate cycle allows the cell to incorporate two-carbon compounds, such as ethanol and acetate, into the TCA cycle and is generally required in *Candida* spp. when its virulence is increased [35]; upon the inhibition of its respiratory chain, *C. albicans* can activate alternative respiratory pathways with different oxidases, which are expressed even in the presence of oxidants, to protect its cells against oxidative stress through reduction of ubiquinone [72,73,74]. However, some studies indicate that the partial inhibition of the electron transport chain may lead to enhanced oxidative stress [75,76,77]. The enhanced levels of malonic acid may also be related to the modified beta-oxidation pathway observed in *C. albicans* and involve the metabolism of propionyl-CoA, a toxic substance. Additionally, malonic acid levels can be enhanced by beta-alanine metabolism [78].

We believe that the upregulation of amino acid metabolism pathways and the concomitant reduction in amino acid concentrations and enhanced levels of byproducts are due to the decreased cellular respiration commonly observed in mature biofilms. DMSO exacerbates the hypoxic and fermentative conditions in the deeper layers of mature biofilms. We postulate that DMSO enhances malonic acid levels, inhibiting the TCA cycle and respiratory chain. Subsequently, we hypothesise that the fermentation process and degradation of amino acids are enhanced to produce energy, enhancing reactive oxygen species (ROS) levels. Additionally, we assume that the enhanced levels of fumaric acid and ketoglutaric acid result from the urea cycle’s higher activity and glutamic acid degradation, respectively, and not from the upregulation of the TCA cycle.

### 4.4. Cofactors and Vitamins Metabolism

Eukaryotic cells rapidly produce mRNAs that encode detoxification and repair proteins during oxidative stress. In *C. albicans*, these detoxification and repair proteins include catalase, glutathione peroxidase, and superoxide dismutase. Additionally, genes encoding components of the thioredoxin and glutathione/glutaredoxin systems are activated in *C. albicans* during oxidative stress. In *C. albicans*, three signalling pathways are activated in response to ROS: Cap1 transcription factor, Hog1 stress-activated protein kinase, and Rad53 DNA damage control kinase [79].

Guedouari et al. [80] demonstrated that glutathione, an important intracellular antioxidant, is consumed during filamentation. The enhanced levels of glutathione and pyroglutamic acid (a precursor) in the DMSO group suggested that DMSO cannot stimulate a morphological transition but can induce oxidative stress. During oxidative stress, protein synthesis is rapidly inhibited in *C. albicans* [81], which was observed in this study’s biofilms from the DMSO group. However, Abegg et al. [82] reported that the intracellular levels of glutathione alone must not serve as a parameter for assessing the response of *Candida* species to oxidative stress.

Ubiquinone-9 (CoQ9) is an isoprenoid quinone found in the organelle membranes of *C. albicans*. It participates in the electron transport chain and plays an important role in protecting its cells against oxidative stress [83,84,85,86]. The DMSO group exhibited an upregulated ubiquinone biosynthetic pathway, which may be a response to oxidative stress or may be due to the activation of alternative respiratory pathways.

Previous studies have reported that DMSO’s cryoprotectant activity is mistakenly attributed to its antioxidant capacity [87,88]. Additionally, Sadowska-Bartosz et al. [89] demonstrated that DMSO enhances oxidative stress and dose-dependently enhances the glutathione levels in *S. cerevisiae*, effects which were also observed in this study.

*Candida albicans* is a biotin auxotroph. Biotin is a cofactor of fatty acid and carbohydrate metabolism [90]. Hussin et al. [91] reported that in addition to serving as an enzymatic cofactor, biotin is a morphological regulator that stimulates germ tube formation and, subsequently, filamentation in *C. albicans*.

Consistent with our findings, Zhang et al. [26] reported that DMSO upregulates biotin metabolism in *S. cerevisiae* but also causes the overexpression of genes involved in the biosynthesis of pyridoxal phosphate, which is involved in the metabolism of vitamin B6. However, in this study, 0.03% DMSO probably downregulated the expression of these genes in *C. albicans* SC5314. The recovery pathway of vitamin B6, which is supplemented in the culture medium with pyridoxine hydrochloride, is responsible for releasing hydrogen peroxide for the synthesis of pyridoxal-5’-phosphate (vitamin B6) from pyridoxine-5’-phosphate. Therefore, we postulated that the vitamin B6 recovery pathway is downregulated in response to the DMSO-induced oxidative stress in the cells. This is because the ROS generated during the recovery would further contribute to cellular damage.

DMSO enhanced the levels of dehydroascorbic acid, a precursor of ascorbate. This indicated the downregulation of the ascorbate and aldarate metabolism pathway, possibly due to the downregulation of carbohydrate metabolism pathways. The oxidation of ascorbate within the cell results in the production of dehydroascorbic acid. However, this study added neither ascorbate nor its derivatives to the culture medium. Surprisingly, previous studies indicate that *C. albicans* does not synthesise ascorbate but synthesises erythroascorbate (dehydro-D-arabinone-1.4-lactone), an analogous ascorbate molecule [92,93,94,95]. Further studies are needed to verify the presence and metabolism of this vitamin in *C. albicans*. Indeed, such studies should check the possibility of spontaneous or DMSO-induced conversion into its reduced form.

### 4.5. Energy Metabolism

Sulphur metabolism is well understood in *S. cerevisiae*. Sulphur metabolism involves the assimilation of inorganic sulphate, which is necessary to synthesise sulphur-containing amino acids. The supplementation of sulphur sources results in cellular stress in Yarrowia lipolytica, due to the production of sulphites, and a concomitant decrease in the levels of glutathione pathway intermediates [96]. This did not concur with the results obtained in *C. albicans* SC5314 treated with DMSO. However, an evolutionary analysis of the pathways of this metabolism in the Saccharomycetes class (*S. cerevisiae*, *C. glabrata,* and *Y. lipolytica*) revealed that there are differences in the groups of enzymes that metabolise sulphur-containing amino acids [97].

According to our observations, DMSO upregulated sulphur metabolism, which may be through the conversion of DMSO into DMS and its subsequent conversion to S-adenosyl methionine. The DMSO group exhibited enhanced levels of S-adenosyl methionine.

*Candida boidinii* is the only methylotrophic *Candida* species. Methylotrophic organisms can use methanol as their sole carbon source. *Candida boidinii* plays an indispensable role in the global carbon cycle, and specifically the methane cycle. It mediates the biotransformation of methane to carbon dioxide [98].

This study observed that DMSO downregulates methane metabolism in *C. albicans* biofilms. Although methane metabolism has not been studied in *C. albicans*, some metabolic reactions may occur due to the presence of enzymes common to metabolic processes. We believe that the enhanced levels of glutathione induced by DMSO may result in the downregulation of methane metabolism. Glutathione is involved in the oxidation of formaldehyde to CO_2_ [99]. *Candida albicans* cannot metabolise methanol to formaldehyde, which concurred with the observations of this study. Formaldehyde can only be obtained from the external environment. However, Klein et al. [100] detected the formation of formaldehyde in biological systems after a reaction between DMSO and hydroxyl radicals (OH^−^). This may explain the reason that we did not detect formaldehyde in this study, as formaldehyde may have been catabolised to CO_2_ or evaporated during lyophilisation.

### 4.6. Secondary Metabolism

Tyrosol and farnesol are the two secondary metabolites produced by *C. albicans*, which are involved in quorum-sensing and inhibiting filamentation. Indol-3-acetic acid is a signalling molecule that has already been reported in *C. albicans*. Indol-3-acetic acid is reportedly involved in filamentation induction and substrate adherence [101]. Although tyrosol was detected in this study, there was no significant difference in the tyrosol levels between the experimental groups.

Betalains are pigments found in plants belonging to the Caryophyllales order and in some higher fungi, such as *Amanita muscaria* [102]. However, betalains are not reported to be present in *C. albicans*. The upregulation of the betalain biosynthetic pathway observed in this study may be due to the standard processes shared among species that cannot generate the same final product.

### 4.7. Final Considerations

*Candida albicans* and DMSO are essential for biological research. However, there are limited studies on the effect of DMSO on the metabolic pathways of *C. albicans*. Most studies have evaluated the effect of DMSO on *S. cerevisiae*. However, these studies are focused on genetic and not metabolic analyses. The phenotype does not always correspond to the genotype of an organism. Hence, post-genomic analyses are essential to understanding cellular physiology [29]. This study is the first to report the effect of DMSO on the metabolic profile of *C. albicans* biofilms.

Our analysis of the metabolome and the possible pathways involved in the intracellular metabolism of *C. albicans* biofilms grown in the absence and presence of DMSO revealed that DMSO affects the metabolism of lipids, the central metabolism of carbon and carbohydrates, the metabolism of amino acids, and cofactors/vitamins. Specifically, DMSO targeted the biosynthesis of fatty acids. DMSO downregulated fatty acid biosynthesis and consequently affected the other metabolic pathways in *C. albicans*, which validated the results obtained in *S. cerevisiae* studies. Additionally, it was postulated that DMSO enhances the oxidative stress in the cells, which also concurred with the results of previous studies [35,89].

For the first time, the presence of a dehydroascorbic acid metabolite, a precursor and product of ascorbate, was identified in *C. albicans*, which must be further investigated.

DMSO alters the metabolic profile of *C. albicans*, so care must be taken when using DMSO as a solvent in drug research. DMSO not only facilitates the penetration of drugs into cells, as reported previously, but also affects cellular metabolism. Additionally, the synergistic effects of DMSO and drugs may affect the validity of drug susceptibility results and thus these drugs cannot be accurately translated into clinical practice.

## Figures and Tables

**Figure 1 jof-10-00337-f001:**
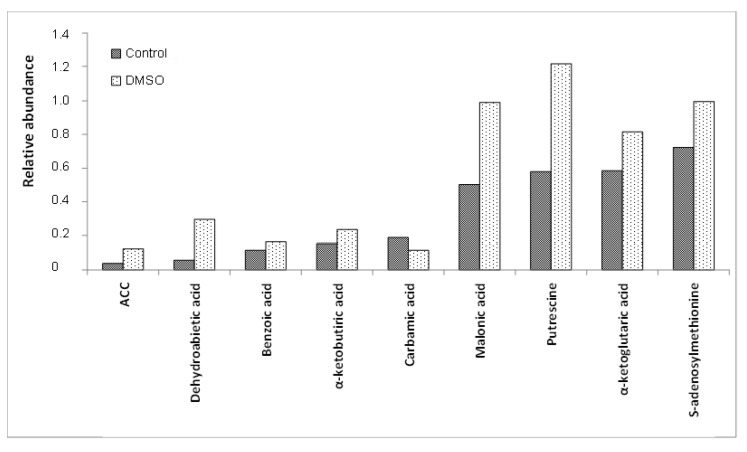
Means of the relative abundances (<1.4) of the *C. albicans* SC5314 intracellular metabolites which showed statistically significant differences (*p* ≤ 0.05) between the control and DMSO groups. ACC = 1-aminocyclopropane-1-carboxylic acid.

**Figure 2 jof-10-00337-f002:**
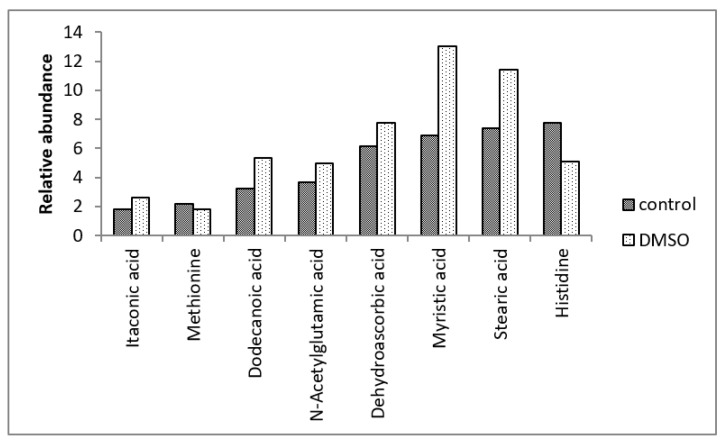
Means of the relative abundances (<14) of the *C. albicans* SC5314 intracellular metabolites which showed statistically significant differences (*p* ≤ 0.05) between the control and DMSO groups.

**Figure 3 jof-10-00337-f003:**
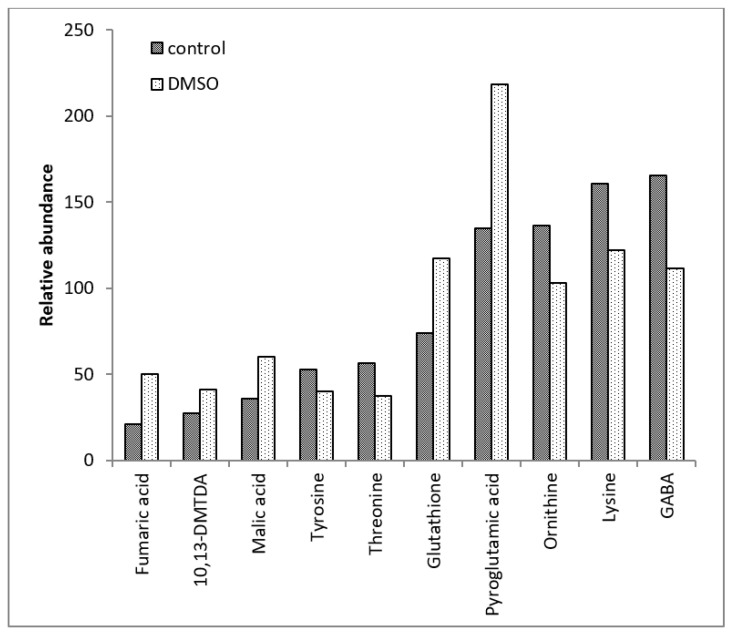
Means of the relative abundances (<250) of the *C. albicans* SC5314 intracellular metabolites which showed statistically significant differences (*p* ≤ 0.05) between the control and DMSO groups. 10,13-DMTDA = 10,13-dimethylthetradecanoic acid; GABA = 4-aminobutyric acid.

**Figure 4 jof-10-00337-f004:**
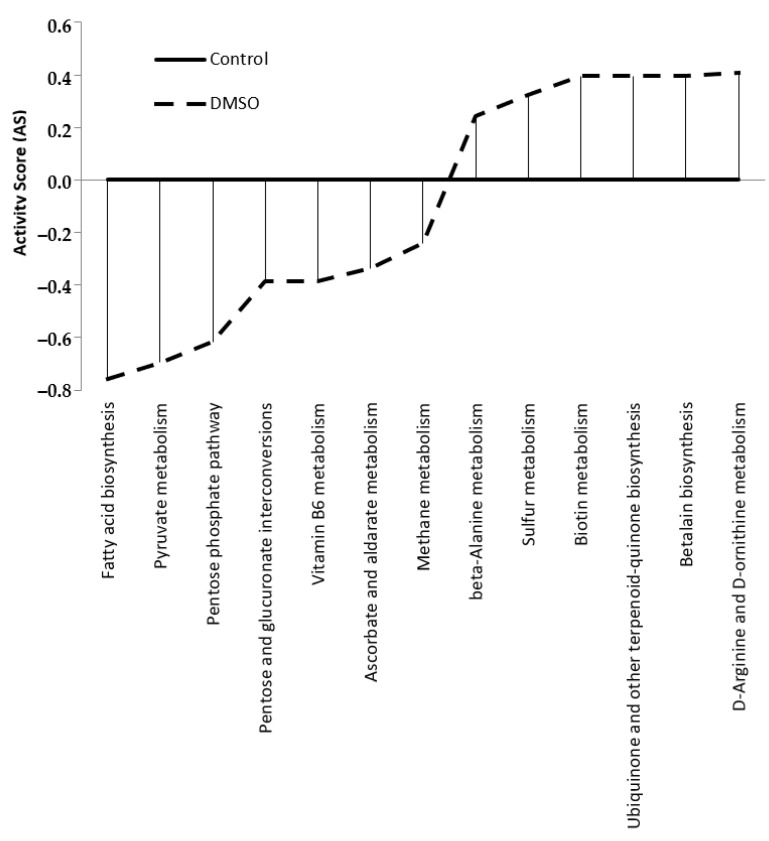
Metabolic pathways with their activity altered (*p* ≤ 0.05) by the presence of 0.03% (*v*/*v*) DMSO in *C. albicans* SC5314 biofilm samples. Pathways with an activity score (AS) less than 0 (zero) represent those downregulated by DMSO, and pathways with an AS greater than 0 (zero) represent those upregulated.

**Scheme 1 jof-10-00337-sch001:**
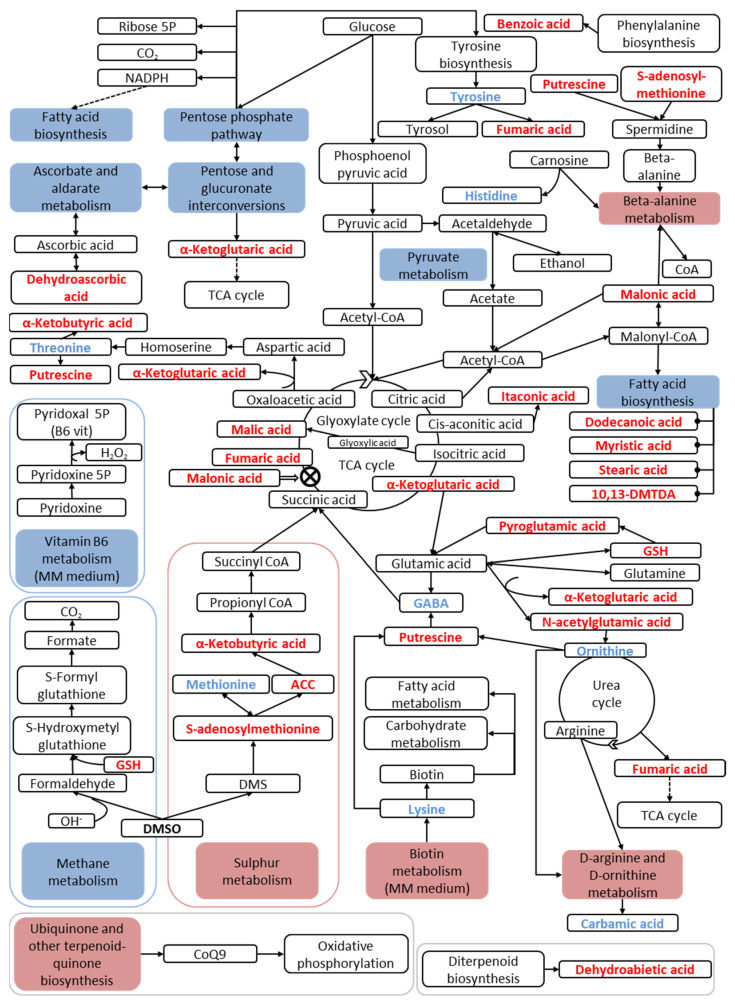
Proposed shifts in the metabolic network of *C. albicans* SC5314 altered by 0.03% (*v*/*v*) DMSO and assessed by GC/MS. Upregulated pathways and increased metabolites are represented in red, and downregulated pathways and reduced metabolites are depicted in blue. ⊗ = inhibition.

**Table 1 jof-10-00337-t001:** Intracellular metabolites identified in mature *C. albicans* SC5314 biofilms in both experimental groups [control and treated with 0.03% (*v*/*v*) DMSO].

Classification of Metabolites	N°	Intracellular Metabolites
Amino acids and their isoforms	22	Alanine, asparagine, aspartic acid, beta-alanine, cysteine, glutamic acid, glutamine, glycine, histidine, isoleucine, leucine, lysine, methionine, norvaline, ornithine, phenylalanine, proline, serine, threonine, tryptophan, tyrosine, valine
Amino acid derivatives	6	Creatinine, cystathionine, N-acetylglutamic acid, putrescine, pyroglutamic acid, S-adenosylmethionine
TCA cycle intermediates	8	Alpha-ketoglutaric acid, alpha-ketobutyric acid, cis-aconitic acid, citric acid, fumaric acid, isocitric acid, malic acid, succinic acid
Fatty acids	17	10,13-dimethyl tetradecanoic acid (10,13-DMTDA), 9-heptadecenoic acid, adipic acid, caprinoic acid, caprylic acid, decanoic acid, docosanoic acid, dodecanoic acid, eicosanoic acid, gamma-linoleic acid, hexanoic acid, myristic acid, palmitelaidic acid, pentadecanoic acid, stearic acid, trans-vaccenic
Glycolytic intermediates	2	2-phosphoenolpyruvic acid, pyruvic acid
Cofactors and vitamins	6	Dehydroascorbic acid, gamma-aminobutyric acid (GABA), glutathione, NADP/NADPH, nicotinamide, nicotinic acid
Others	27	1-aminocyclopropane-1-carboxylic acid (ACC), 2-aminoadipic acid, 2-hydroxybutyric acid, 2-isopropylmalic acid, 3-methyl-2-oxopentanoic acid, 4-aminobenzoic acid, 4-hydroxyphenylacetic acid, 4-hydroxyphenylethanol (tyrosol), 5-oxotetrahydrofuran-2- benzoic acid, carbamic acid, carboxylic acid, citramalic acid, dehydroabietic acid, dibutylphthalate (DBP), DL-hydroxyglutaramate, dodecane, EDTA, glutaric acid, glyceric acid, glyoxylic acid, itaconic acid, lactic acid, levulinic acid, malonic acid, oxalic acid, para-toluic acid, quinic acid
Total no. of identified metabolites	88	

**Table 2 jof-10-00337-t002:** Intracellular metabolites of mature *C. albicans* SC5314 biofilms altered by 0.03% (*v*/*v*) DMSO.

Classification of Metabolites	Intracellular Metabolites Altered by DMSO	
	Upregulated	Downregulated
Amino acids and their isoforms	-	Histidine, lysine, methionine, ornithine, threonine, tyrosine
Amino acid derivatives	N-acetylglutamic acid, putrescine, pyroglutamic acid, S-adenosylmethionine	-
TCA cycle intermediates	Alpha-ketobutyric acid, alpha-ketoglutaric acid, fumaric acid, malic acid	-
Fatty acids	10,13-dimethyl tetradecanoic acid (10,13-DMTDA), dodecanoic acid, myristic acid, stearic acid	-
Cofactors and vitamins	Dehydroascorbic acid, glutathione	Gamma-aminobutyric acid (GABA)
Others	1-aminocyclopropane-1-carboxylic acid (ACC), benzoic acid, dehydroabietic acid, itaconic acid, malonic acid	Carbamic acid
Total no. of altered metabolites	19	8

**Table 3 jof-10-00337-t003:** Metabolic pathways with their activity altered by 0.03% (*v*/*v*) DMSO in *C. albicans* SC5314 biofilm samples.

Metabolic Pathway Groups	Metabolic Pathways Altered by DMSO	
	Upregulated	Downregulated
Amino acid metabolism	Beta-Alanine metabolism D-arginine and D-ornithine metabolism	-
Lipid metabolism	-	Fatty acid biosynthesis
Carbohydrate metabolism	-	Pentose phosphate pathway Pentose and glucuronate interconversions Ascorbate and aldarate metabolism
Carbon metabolism	-	Pyruvate metabolism
Energy metabolism	Sulphur metabolism	Methane metabolism
Cofactor and vitamin metabolism	Biotin metabolism Ubiquinone and other terpene-quinones biosynthesis	Vitamin B6 Metabolism
Secondary metabolites metabolism	Betalain biosynthesis	-
Total	6	7

## Data Availability

No new data were created or analyzed in this study. Data sharing is not applicable to this article.

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
