# Peer review of "Central Carbon Metabolism in Candida albicans Biofilms Is Altered by Dimethyl Sulfoxide"

_jof, 2024, doi:10.3390/jof10050337_

Round 1

Reviewer 1 Report

The manuscript submitted by Arruda et al. investigates the effect of DMSO on the metabolism of Candida albicans biofilms made under a continuous flow system. Despite the enormous literature published on this topic related to many different microorganisms, the authors claimed their work is significant due to the differential abundance of metabolites uniquely identified in this study. The overall conclusion of the work is somewhat common, as there have been discussions on the use of DMSO for antimicrobial drug testing. Overall, the study is interesting with the identification of unique metabolites (but may need additional justification so they can be highlighted) however, these observations are likely highly context dependent therefore authors should be careful in interpreting their results for generalized conclusions.

Despite many available in situ and flow-induced biofilm formation techniques, authors did not justify the use of PEBR system, particularly with paper discs. The authors should provide evidence for the use of higher flow rate and longer incubation conditions (24 hrs is the most common incubation time). The authors have not given any evidence for successful biofilm formation using the PEBR system under the continuous fluid flow on paper discs. The authors could have quantified their biofilms under two treatments to show that fluid flow provides optimum biofilm formation with or without DMSO treatment. 

The authors used paper discs (organic material) rather than an insert material as the substratum for biofilm formation. Therefore, the authors should justify or provide evidence for the neutral impact of paper discs (compatibility of the paper discs with DMSO) upon DMSO treatment and metabolomic analysis. The authors did not mention how paper discs were removed when extracting the biofilm for GC/MS analysis.  

For the biofilm formation, this study has used an MM medium, presumably minimal medium which is not a well supportive medium for C. albicans biofilm formation. I suggest the authors quantify biofilm formation in this particular media with reference to a standard Spider or RPMI media using the same system to ensure abrupt biofilm formation under the given restricted nutritional environment. 

It is well known that microorganisms responds to DMSO differently particularly at very low and intermediate concentrations, depending on the species, specific strain, metabolic state and growth conditions. The authors attempted to explain their findings mainly focusing on metabolic aspects however, the main possible reason for contrasting observation likely to be the unique biofilm formation conditions used by the authors. Shear stress could be another main reason for technical difficulties (such as harvesting fully mature biofilm). Authors should refer to previous literature on how increased share stress reduces biofilm growth and increases metabolic rates. 

There are no details on the biological or technical replicates used for GC/MS analysis. Are the analysis based on composite samples?

The authors should discussed the advantages or disadvantages of the study methods (effectiveness, efficiency or sensitivity, etc.,) which could impact the biofilm formation or metabolomics.

1. The discussion is unnessarily lengthy and followed a review article format. The issue is the authors are trying bring every aspects of metabolism to justify their data without screening for the relevant literature. The introduction could have been strengthen by previous literature which have been included in the discussion as of now.

2. Table 1 could have been presented as a dot plot or volcano plot for better visualization. 

3. The term 'abundance' does not necessarily mean 'upregulation.' For example, line 463. Did the authors test for the expression of these genes?

4. Fig 1-3 could be a figure panel rather than 3 separate figures. 

5. The composition of the MM medium should be included as this is a metabolomic study. The upregulation or downregulation of metabolism should be explained in terms of the growth media as well.

6. Scheme 1: The legend should be descriptive enough to briefly include the methodology. What is 'Metionina'? The image as of now is somewhat blurry. 

 7. There are certain instances where the references are required for the discussion section. Eg. 431-435, 367-368, 382-384, 539-540,531-534, 509-511 etc,.

8. Interestingly authors found the presence of dehydroascorbic acid while there is no sources for ascorbic acid. Yeast produces erythro-ascorbic acid instead of ascorbic acid and authors should check the possibility of spontaneous conversion in to its reduced form as this is a significant finding of their work. 

Author Response

Response to Reviewer #1

Initially, the authors would like to thank the reviewers for the time spent reading and asking questions that enriched the text, making it more understandable.

All points raised were reviewed and rethought; when considered relevant, they were corrected.

In response to reviewer #1, we understand that our results show that DMSO affects several metabolic pathways. Our conclusion may seem generic when we argue that its use must be considered carefully, but this results from simultaneous action in several ways.

Another publication (Selow et al., 2015) extensively presented the use of PEBR, with numerous technical details and methodological validity. The continuous feeding of biofilms in this system guarantees cellular vitality superior to that of static biofilms grown on culture plates. We analyzed 72-hour biofilms because, as seen in several other studies, these biofilms are mature at this age. DMSO does not establish any interaction with the cellulosic matrix of the discs. Details of the removal of biofilms from discs have been added to the text.

Details of the liquid culture medium have been added. We ensured that there was significant growth of biofilm on the discs. The purpose of the study was not to quantify biofilm growth (which has already been shown in a previous study by Selow et al.).

We believe that the low shear stress to which the biofilms were subjected was not an external condition that caused the metabolic changes.

The issue of replicas was duly corrected.

Articles have been moved from discussion to introduction, as suggested.

We allow ourselves to disagree with the reviewer's opinion regarding Table 1, as we think a volcano plot is unsuitable for presenting the data displayed in the table.

The issue of “upregulation” was resolved, used only for pathways and not genes.

Figures 1, 2, and 3 differ in relation to the quantities presented on the y-axis; therefore, they must be maintained as independent figures.

Scheme 1 has been corrected.

The issue of deoxyascorbic acid had already been discussed in the text.

Reviewer 2 Report

The authors investigated the effect of low concentrations DMSO on the metabolic profile of C. albicans biofilms. This is relevant to many studies of potential novel antifungal drugs, as DMSO is often used as solvent. The manuscript was well written, but can be improved by the following minor recommendations:

1. In the introduction, mention is made of antibiotics in the context of antifungal drugs. Although antifungals are technically also antibiotics, the more common definition of antibiotics refer strictly to certain antibacterial drugs. Please consider changing the term here to antifungal instead.

2. In the materials and methods section the authors mention "normal culture medium" (ln 82) and later MM (please define the abbreviation). Please provide more detail regarding these media.

3. In the results, mention is made of the results of the PCA analyses, but not shown. Please include this - it can be as supplementary material or in the manuscript.

4. For all the tables, please include horizontal lines between the different types of metabolites as it is very difficult to read the tables as they are currently

5. Please refer to all figures in the preceding text (especially figures 1-3).

See my comments above

Author Response

Response to Reviewer #2

Initially, the authors would like to thank the reviewers for the time spent reading and asking questions that enriched the text, making it more understandable.

All points raised were reviewed, rethought and, when considered relevant, corrected.

In response to reviewer #2, all issues were promptly corrected. Regarding the issue of PCA analysis, we understand that its results do not need to be presented, as they are an intermediate part of the metabolomic analysis process and not the final one. In the text, there is information that this data will not be presented.

Reviewer 3 Report

In the manusrcript "Change in the metabolic profile of Candida albicans biofilms induced by dimethyl sulfoxide" the authors described alteranations in fungal metabolic pathways in the presence of one of the commonly used solvent for substances tested in laboratory conditions for their effect on the microorganisms. These are important observations from the point of view of planning the testing of novel antifungal drugs, so they may provide some relevant information for future research. However, the manuscript requires some revisions.

The role of DMSO for Candida and other fungi should be explained in the introduction, instead discussion section.

Please explain why the DMSO concentration of 0.02% has been chosen for analysis, since authors stated in the introduction section that 0.03% is usually used for drug testing?

The methodological part requires rearrangement and specification, e.g. please provide the composition of the MM medium, the part regarding the isolation of metabolites should be arranged differently, it is currently unclear, details of the data analysis methodology should be provided in section 2.10 instead of being referred to another paper.

How does this concentration affect the growth of the biofilm and its overall metabolic activity (measured, for example, by the MTT or XTT test). The impact on membrane permeability and the integrity of the cell wall of cells in the biofilm should also be taken into account and supplemented in results.

How exactly was the relative abundance of metabolites determined, it should be included in the manuscript text.

The metabolomics data should have been deposited in the appropriate public repository, and link included in the manuscript.

Figures 1-3 can be combined into one panel composed of three separate graphs.

The discussion is very extensive, some information may be included also in the introduction section as mentioned above.

Abstract line 17 - it should be sulfoxide instead of sulphoxide, please verify thorought the manuscript text (line 50 - the same).

Line 237 - it should be: "inhibition of formation of germ tubes", or "inhibition of germination".

In Figures 1,2,3 standard deviation SD is not indicated.

The reference list should be formatted as required. 

Author Response

Response to Reviewer #3

Initially, the authors would like to thank the reviewers for the time spent reading and asking questions that enriched the text, making it more understandable.

All points raised were reviewed, rethought and, when considered relevant, corrected.

In response to reviewer #3, all questions were duly corrected.

Articles have been moved from discussion to introduction as suggested.

In fact, the concentration of DMSO used was 0.03%. A typing error spread to different parts of the text. After reviewing the protocols, we verified the error.

The composition of the culture medium was provided.

As the text became long, it was decided to reference the methodology used in relation to GC/MS as a citation since the ipsis litteris methodology was followed.

DMSO did not interfere with the amount of biofilm formed, as mentioned in the text.

Relative abundances are reported following the widely used criterion of comparing the abundance of each compound with that of the highest abundance.

Deposit data is still being considered by the authors.

Figures 1, 2, and 3 differ in relation to the quantities presented on the y-axis; therefore, they must be maintained as independent figures.